# Novel strains of Tomato Spotted Wilt Orthotospovirus (TSWV) are transmitted by western flower thrips in a context-specific manner

Senthilraja Chinnaiah[1,2,3], Arinder K. Arora[2,3], Kiran R. Gadhave [2,3]*

1 School of Agriculture, Middle Tennessee State University, Tennessee, 2 Texas A&M AgriLife Research, Amarillo, Texas, 3 Department of Entomology, Texas A&M University, College Station, Texas

* kiran.gadhave@ag.tamu.edu

## Abstract

Novel resistance breaking (RB) strains of tomato spotted wilt orthotospovirus (TSWV) capable of disrupting single gene resistance in tomato (*Sw-5b*) and pepper (*Tsw*) have been reported worldwide. Thrips, a supervector of TSWV, transmit these strains in a suite of specialty and staple food crops across the globe. However, transmission biology of RB strains remains virtually unexplored. We investigated various transmission parameters *viz.* inoculation efficiency, putative sex-specific differences in inoculation, virus accumulation, and source sink relationships to dissect these interactions. Six novel strains of TSWV, namely Tom-BL1, Tom-BL2, Tom-CA, Tom-MX, Pep-BL and Non-RB, transmitted by western flower thrips (WFT) were used and thrips were allowed four 24h consecutive inoculation accession periods (IAPs). Our results show that most strains were inoculated at all four IAPs, however, their rates differed across IAPs. Overall, WFT had highest inoculation efficiency at the first and lowest at the second IAP. Female thrips carried higher virus titers; however, males were better at inoculating TSWV. Furthermore, we did not find significant positive correlations in virus titers between the tissues used for TSWV acquisition, thrips and thrips-inoculated leaf discs. Males inoculated RB strains at 87% efficiency whereas Non-RB strain at 80% efficiency. Female thrips were 77% and 75% efficient at inoculating RB and Non-RB strains, respectively. This study furnishes new insights into the transmission biology of TSWV RB strains, especially from inoculation and thrips sex perspectives, and provides a baseline for future molecular studies surrounding ever evolving novel TSWV strains.

## Introduction

Tomato spotted wilt orthotospovirus (TSWV), also known as *Orthotospovirus tomato-maculae*, is a type member of the *Orthotospovirus* genus in the family *Tospoviridae*.

**Data availability statement:** All relevant data are within the manuscript and its Supporting Information files.

**Funding:** Texas A&M AgriLife Research Insect Vectored Diseases Grant. The funders had no role in study design, data collection and analysis, decision to publish, or preparation of the manuscript.

**Competing interests:** NO authors have competing interests.

TSWV is recognized as the most widespread plant-infecting RNA virus [1]. It affects more than 1,000 agriculturally valuable crops, including tomatoes and peppers, and causing significant economic losses by reducing both yield and quality of agricultural product [2]. In recent decades, two resistance genes, *Sw-5b* and *Tsw*, have been identified in tomato and pepper, respectively, and utilized against TSWV through resistance breeding strategies [3–6]. Tomato and pepper cultivars carrying these genes exhibited strong and prolonged resistance to TSWV. However, the intensive cultivation of resistant cultivars in recent decades has driven the global emergence of resistance-breaking (RB) strains of TSWV, now reported in the United States, Mexico, Italy, Australia, China, Argentina, Spain, Hungary, South Africa, Turkey, South Korea, and Brazil [7–19].

TSWV is transmitted in a persistent and propagative manner by plant-feeding insect vectors, namely thrips (Thysanoptera: Thripidae) [20]. Among different thrips species, *Frankliniella occidentalis* (Western flower thrip, WTF) is considered as the most effective vector due to its high reproductive rate as well as concealed and polyphagous behavior [21]. After acquisition by the first and early second instar larvae, TSWV replicates in the midgut epithelia and adjacent cells of the midgut intestinal muscles, followed by invasion of the primary salivary glands. Viruliferous adults transmit the virus when they secrete saliva into plants while feeding [21–24]. Although males carry a lower viral titer, they are more efficient vectors than females [25], likely due to their greater mobility, distinct feeding behavior, and minimal leaf scarring [26]. In contrast, sedentary females cause more localized damage, which may obstruct virus movement and reduce transmission efficiency. WFT fails to transmit TSWV when they acquire the virus as adult, because the ingested virus accumulates in amorphous electron dense material in epithelial cells and fails to disseminate to other tissues [22].

TSWV infection has synergistic effects on the *F. occidentalis* fitness as it influences thrips behavior, and increases fecundity, survival, and longevity [27–29], which has been attributed partly to modulating metabolic and plant defense pathways in plants [30]. Most of these studies have used a single isolate of TSWV, however, we recently investigated the WFT fitness after acquisition of four resistance breaking (RB) and one non-resistance breaking ('Non-RB') strain. Our lab in an earlier study found that RB strains significantly increased WFT fitness measured via fecundity (i.e., number of offsprings), and adult period (i.e., first to last day of adulthood) compared to the Non-RB strain and non-viruliferous controls [31]. Furthermore, RB-viruliferous thrips transmitted TSWV more efficiently than the Non-RB strain [31]. The term 'strain' rather than 'isolate' is used to describe our TSWV strains because they cause distinct symptoms in tomato and/or pepper plants (i.e., host phenotype) and have some distinct genetic mutations [32]. This first-ever study on thrips transmission of RB strains suggested that vector-imposed selection pressures, besides single gene resistant hosts, may play an important role in the emergence and spread of new RB strains.

This follow-up study builds on our previous findings to gain deeper insights into the transmission biology of TSWV RB and non-RB strains. Using sequential inoculation

assays—where viruliferous WFT feed on a series of leaf discs—we aim to test whether: (i) viral copy numbers differ across thrips sexes before and after the inoculation access period (IAP), as well as across the four sequentially inoculated leaf discs; (ii) a source-sink relationship exists among virus titers in acquisition leaf tissues, thrips, and thrips-inoculated leaf discs; and (iii) inoculation efficiency of TSWV strains, measured by the percentage of leaf discs infected by male versus female thrips, differs.

## Methods

### *Frankliniella occidentalis* colony maintenance

A colony of western flower thrips, *F. occidentalis* originally obtained from Diane Ullman at UC, Davis was used for transmission experiments. This non-viruliferous WFT colony was maintained on surface-sterilized green bean pods in semi-transparent plastic containers with a lid fixed with insect-proof mesh in the center at 25°C and a 16-h photoperiod as described in our prior study [31].

### Maintenance of TSWV strains

A total of five RB and one Non-RB TSWV strains, isolated and characterized previously, were used in this study [17,19,31,32]. All strains were maintained through a cycle of mechanical and thrips transmission in live host plants in insect-proof cages kept in a greenhouse at 25°C and a 12-h photoperiod. Among the five RB strains, four, namely Tom-BL1, Tom-BL2, Tom-CA, and Tom-MX, were isolated from different sources (infected leaf tissue or fruits) and maintained in a tomato cultivar (cv. Celebrity) carrying the *Sw-5b* gene. Among the four tomato RB strains, Tom-BL1 and Tom-BL2 showed typical symptoms of TSWV such as chlorotic patches, concentric rings, and necrotic spots on leaves, while Tom-CA and MX showed puckering and mosaic molting of leaves. Furthermore, Tom-CA exhibited shoestring symptoms on leaves. The fifth RB strain, Pep-BL, was isolated from pepper and maintained in a cultivar containing the *Tsw* gene (cv. Procraft). The non-RB strain, which is unable to overcome resistance conferred by *Sw-5b* or *Tsw*, was originally isolated from pepper and maintained on the susceptible tomato cultivar 'Hot-Ty'.

### TSWV inoculation

TSWV inoculation experiments comprised of four consecutive IAPs of 24-h were performed using modified procedures from Rotenberg et al [25]. A cohort of ~100 first instar larvae (0–24 hour old) of WFT were allowed to acquire TSWV. For virus acquisition the larvae were given an acquisition access period (AAP) of 72-h on symptomatic leaves from plants (tomato or pepper) infected with TSWV in a petri dish as described by Chinnaiah et al [31] (Fig 1). Six such cohorts were used to acquire one strain each.

Following AAP, thrips were kept on sterilized clean bean pods until adulthood. On second day after adult-eclosion, total of 10 individual female and male (n = 10 each/strain) thrips, separated based morphometric characters [33], were transferred to a sterile 2 ml centrifuge tube (1 thrip/tube) containing leaf disc (1 × 1 cm) of either resistant tomato (cv. Celebrity [for Tom strains]) or pepper (Cv. Procraft [for Pep-BL strain]) or susceptible tomato (cv. Hot-Ty [for Non-RB strain]) and allowed to inoculate the virus for 24-h. A small piece (2 × 2 cm) of tissue paper was placed in the centrifuge tube to absorb moisture. The leaf discs were used one per thrips and each IAP, resulting in a total of 10 replicates/sex/IAP/strain for a total of 20 leaf discs/IAPs/strains. After 24-h of inoculation access period (IAP), leaf discs were replaced with fresh leaf discs to start the next 24 hours IAP, and inoculated leaf discs were floated on sterile nucleus free water for 48-h in a 96 well microtiter plate with an appropriate space between the samples. After the incubation, water from the leaf disc was wiped with sterile Kim wipes and leaf discs were stored at −80ºC until the extraction of RNA. Likewise, four consecutive IAPs (IAP-I, II, III, and IV) were performed for all the six strains simultaneously (Fig 1).

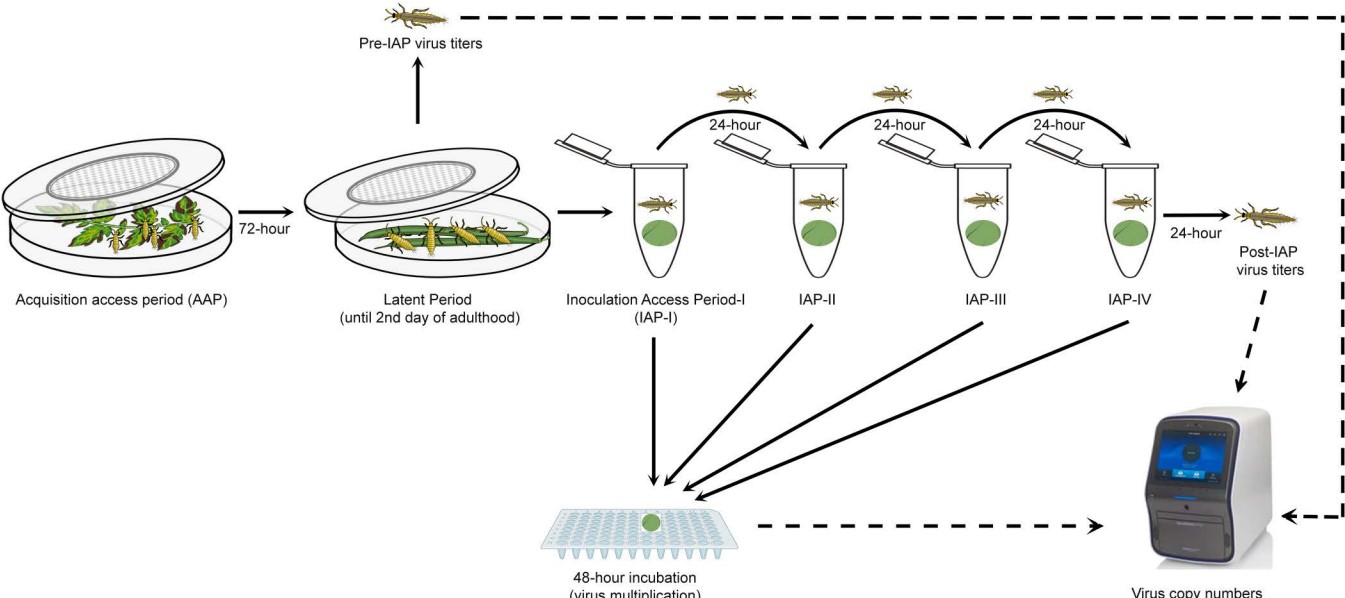

**Fig 1. Visual depiction of methods used in sequential inoculation of resistance breaking (RB) and Non-RB TSWV strains by *Frankliniella occidentalis*.** Six cohorts each of approximately, ~100, 0-24 h old larvae were allowed a 72–h acquisition access period on symptomatic leaves infected with one of each of five RB strains and one Non-RB strain of TSWV, separately in a transparent petri-dish covered with insect mesh in the center. After 72h AAP, thrips were reared on surface sterilized green beans until adulthood. On second day after adult-eclosion, individual female and male (n = 10 each sex/strain) thrips were transferred to a sterile 2 ml centrifuge tube containing leaf disc (1 × 1 cm), separately and allowed to inoculate the virus for 24 h. Strain specific tomato or pepper leaf discs from TSWV negative plants confirmed via qRT-PCR were used. After 24 hours leaf disc (1 × 1 cm) was replaced, and insects were allowed a second IAP of 24 hours. Insects were given 4 consecutive IAPs to inoculate virus, successively. After each IAP, inoculated leaf discs were incubated on sterile water for 48- h in 96 well microtiter plate. Following incubation, TSWV copy number from inoculated leave discs were quantified through RT-qPCR. Further, TSWV copies was quantified from a subset of adults prior to IAP (n = 10/sex/strain) and post-IAP (from the individual insects used for consecutive inoculations event).

## RNA extraction and TSWV quantification

Total RNA from individual leaf discs was extracted using Tri-Reagent (Thermo Fisher Scientific, Waltham, MA, USA) as per manufacture's protocol and stored at −80°C until further use. Total RNA from individual female and male thrips (n = 10 for each sex) was extracted immediately after acquisition and completion of four consecutive IAPs using Quick Extract solution (Lucigen, Middleton, WI) as described by Chinnaiah et al [31].

A one-step RT-qPCR was performed using 20-μl reaction mixture containing 5-μl of TaqMan Fast Virus 1-Step Master Mix [4X] (Thermo Fisher Scientific, Waltham, MA, United States), 1-μl probe [20X] ([6~FAM] CAGTGGCTCCAATCCT[BH-Q1a~Q]) and 1-μl primer pair [10 pmol of each] (5′-AGAGCATAATGAAGGTTATTAAGCAAAGTGA-3′) and (5′-GCCT GACCCTGATCAAGCTATC-3′) targeting nucleocapsid (N) gene using QuantStudio 7 Pro system (Applied Biosystems, Waltham, MA, United States) with the following conditions: 50°C for 10 min, holding at 94°C for 5 min, followed by 40 cycles of 94°C for 10 s and 60°C for 30 s [31]. One ng RNA was used in each RT-qPCR to maintain the comparability between treatments. Virus copy numbers/ng of RNA were estimated using standard curve generated by known copy of tenfold serially diluted plasmid containing TSWV N gene [31].

## Statistical analysis

All experimental data were analyzed using R version 3.6.0 [34]. Shapiro-wilk test was performed for virus copy number data to assess the assumption of normality and homogeneity [35]. Since the virus copy number data did not conform to

normality, and was over-dispersed, a generalized linear mixed model (GLMM) from "glmmTMB" package [36] was used for viral copy analysis across different treatment groups. The best fit model was chosen based on the lowest AIC value to avoid overfitting. IAP wise, virus copy numbers in leaf discs were analyzed both separately by female and male thrips and combining copy number across different strains. Virus copy numbers of RB strains were compared with Non-RB strain using post-hoc Dunnett test using "DescTools" package [37]. Virus copy number in leaf disc inoculated by males and females within the strains was compared using student's *t*-test. Furthermore, virus copies in leaf disc between four IAPs within the strain and between sexes were also analyzed using GLMM and treatment means were compared through post-hoc Tukey test using "multcomp" package [38]. A regression analysis was performed to determine a relationship between TSWV copy numbers in thrips, infected tissues, and thrips-inoculated leaf discs. Percent inoculation efficiency was analyzed using generalized linear mode (glm) followed by post-hoc Tukey test.

## Results

### Inoculation of different TSWV strains by WFT

**Tom-BL1.** Both male and female thrips inoculated significantly higher copies of Tom-BL1 in IAP-I compared to IAP-II, III, and IV ($P<0.001$; Fig 2a). Female thrips inoculated lowest virus numbers in IAP-II and males inoculated the lowest during IAP-III, followed by a slight increase in the IAP-IV for both sexes (Fig 2a).

Further, strain Tom-BL1 copy numbers inoculated by both females and males in IAP-I were significantly higher when compared to Non-RB copy numbers inoculated by females ($P<0.001$; S1a Fig.) or males ($P<0.001$; S1b Fig). When inoculation by males and females was combined together, Tom-BL1 titers were inoculated in significantly higher number

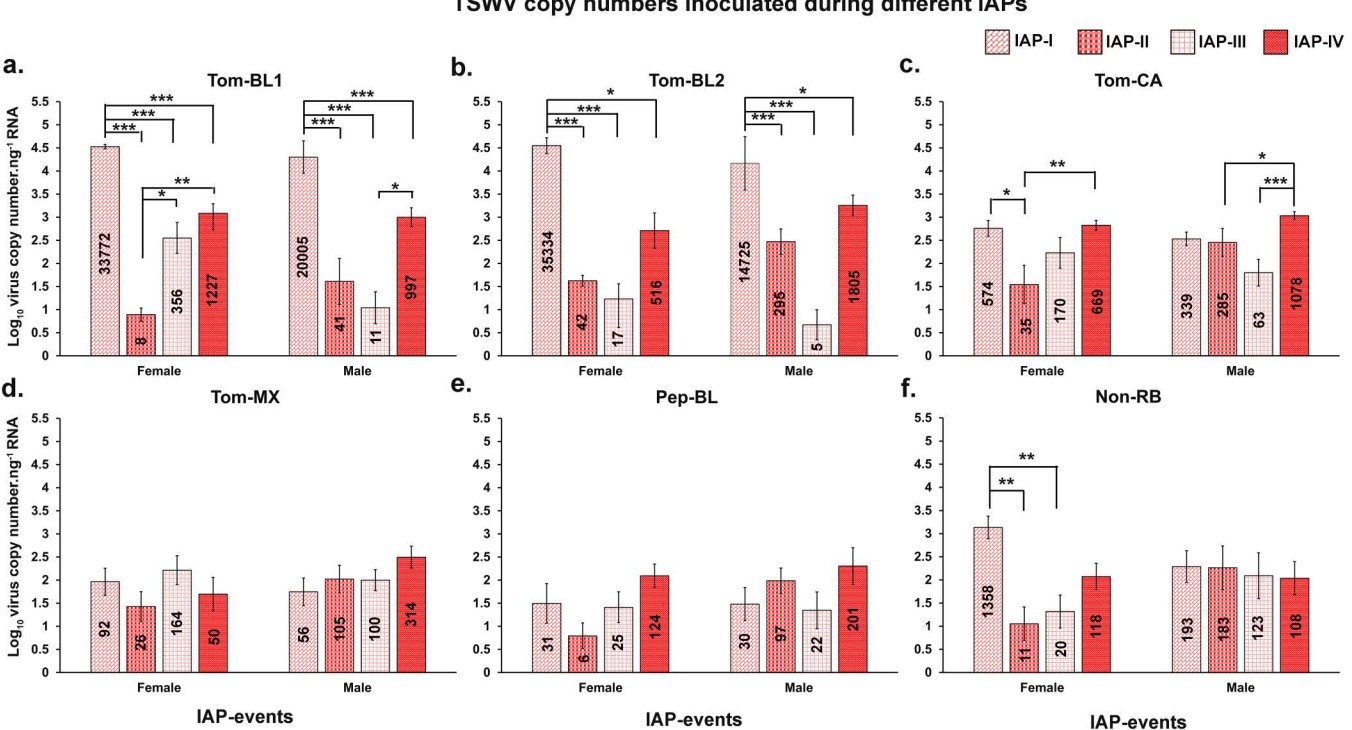

**Fig 2. Comparison of TSWV copies.ng⁻¹ of RNA inoculated by *Frankliniella occidentalis* between IAPs for each strain.** Average copy number of different strains inoculated by female or male *F. occidentalis* (a) Tom-BL1; (b) Tom-BL2; (c) Tom-CA; (d) Tom-MX; (e) Pep-BL, and (f) Non-RB strains. Asterisks indicate significant differences at α = 0.05 (*$P<0.05$, ** $P<0.01$, *** $P<0.001$).

during IAP-I ($P<0.001$; S1c Fig) and IAP-IV ($P=0.004$; S4c Fig) compared to Non-RB. A significant difference in Tom-BL1 copy number inoculated by females as compared to males was observed only in IAP-III ($P=0.015$; S3d Fig).

**Tom-BL2.** Virus copy number of Tom-BL2 inoculated by females and males was significantly higher in IAP-I than other IAP-I & II ($P<0.001$; Fig 2b) and IAP-III ($P<0.05$; Fig 2b). The lowest virus copies were inoculated at IAP-III and then increased slightly at IAP-IV for both females and males.

Furthermore, strain Tom-BL2 was inoculated in significantly higher copies by both females and males than Non-RB females ($P<0.001$, $P<0.05$; S1a Fig) or males ($P<0.001$; S1b Fig) in IAP-I. Copy numbers of Tom-BL2 inoculated by males were significantly higher as compared to Non-RB copy numbers inoculated by females in IAP-II ($P=0.023$; S2a Fig) and males in IAP-IV ($P=0.040$; S4a Fig) IV, respectively. In addition, Tom-BL2 was inoculated in significantly higher numbers than Non-RB when *F. occidentalis* sex was disregarded in IAP-I ($P<0.001$; S1c Fig) and IAP-IV ($P=0.004$; S4c Fig). When inoculation was compared between sexes, females inoculated Tom-BL2 at a higher copy number than males in IAP-I ($P=0.042$; S1d Fig).

**Tom-CA.** For Tom-CA strains, significantly different number of virus copies were inoculated between the IAPs by both female and male (Fig 2c). The female in IAP-II inoculated the lowest virus copies and was significantly different from first and fourth IAP ($P<0.05$, $P<0.01$; Fig 2c), whereas male inoculated the lowest copy numbers at IAP-III and was statistically different from IAP-IV ($P<0.001$; Fig 2c).

No significant difference in Tom-CA copy numbers was observed from Non-RB females and males in any of the IAPs (S1, S2, S3, S4 Fig). However, Tom-CA was inoculated in significantly higher numbers than Non-RB strain when *F. occidentalis* sex was disregarded in IAP-IV ($P=0.004$; S4c Fig)

**Tom-MX.** For the strain Tom-MX, lowest copy numbers were inoculated during second and first IAPs for females and males, respectively. However, no statistical difference was found between IAPs in females ($P=0.363$; Fig 2d) and males ($P=0.384$; Fig 2d).

No significant differences were found between viral copy numbers inoculated by males and females with Non-RB strain (S1, S2, S3, S4 Fig). Additionally, comparison between virus copy numbers inoculated by males and females did not show any significant difference in all the IAPs (S1d, S2d, S3d, S4d Fig).

**Pep-BL.** For the Pep-BL strain, lowest copy numbers were observed during second and third IAPs for females and males, respectively. However, no statistical difference was found between IAPs of females ($P=0.067$) or males ($P=0.384$; Fig 2e).

Pep-BL, however, was inoculated at significantly lower copy numbers by males than Non-RB strain inoculated by females in IAP-I ($P<0.05$; S1a Fig). In addition, Pep-BL copy numbers inoculated by males were significantly higher than those inoculated by females of same strain in IAP-II ($P=0.017$; S2d Fig).

**Non-RB.** For the Non-RB strain, female inoculated significantly higher virus copies at the first IAP compared to second and third IAPs ($P<0.01$; Fig 2f). No significant difference was found in copy numbers inoculated by males between the IAPs ($P=0.912$; Fig 2f).

Further Non-RB copy numbers inoculated by males were significantly higher than females only in IAP-II ($P=0.047$; S2d Fig).

## TSWV copies in adult thrips

**Pre-IAP.** When compared to Non-RB strain copy numbers in infected female thrips, the males carried lower copies of Tom-CA ($P=0.016$) and Pep-BL strain ($P=0.028$; Fig 3a), and males carried higher copies of Tom-BL2 strain compared to Non-RB strain copies in male ($P<0.001$; Fig 3b). No significant difference was observed when TSWV copies of other RB strains carried by males or female were compared to copies of Non-RB strain carried by males or females (Figs 3a & 3b). When data were combined for males and females, the virus copies of Tom-BL2 were the highest in *F. occidentalis* adults and therefore significantly different from Non-RB strain copies carried by thrips ($P=0.05$, Fig 3c). No statistical difference was observed between males and females for any strain (Fig 3d).

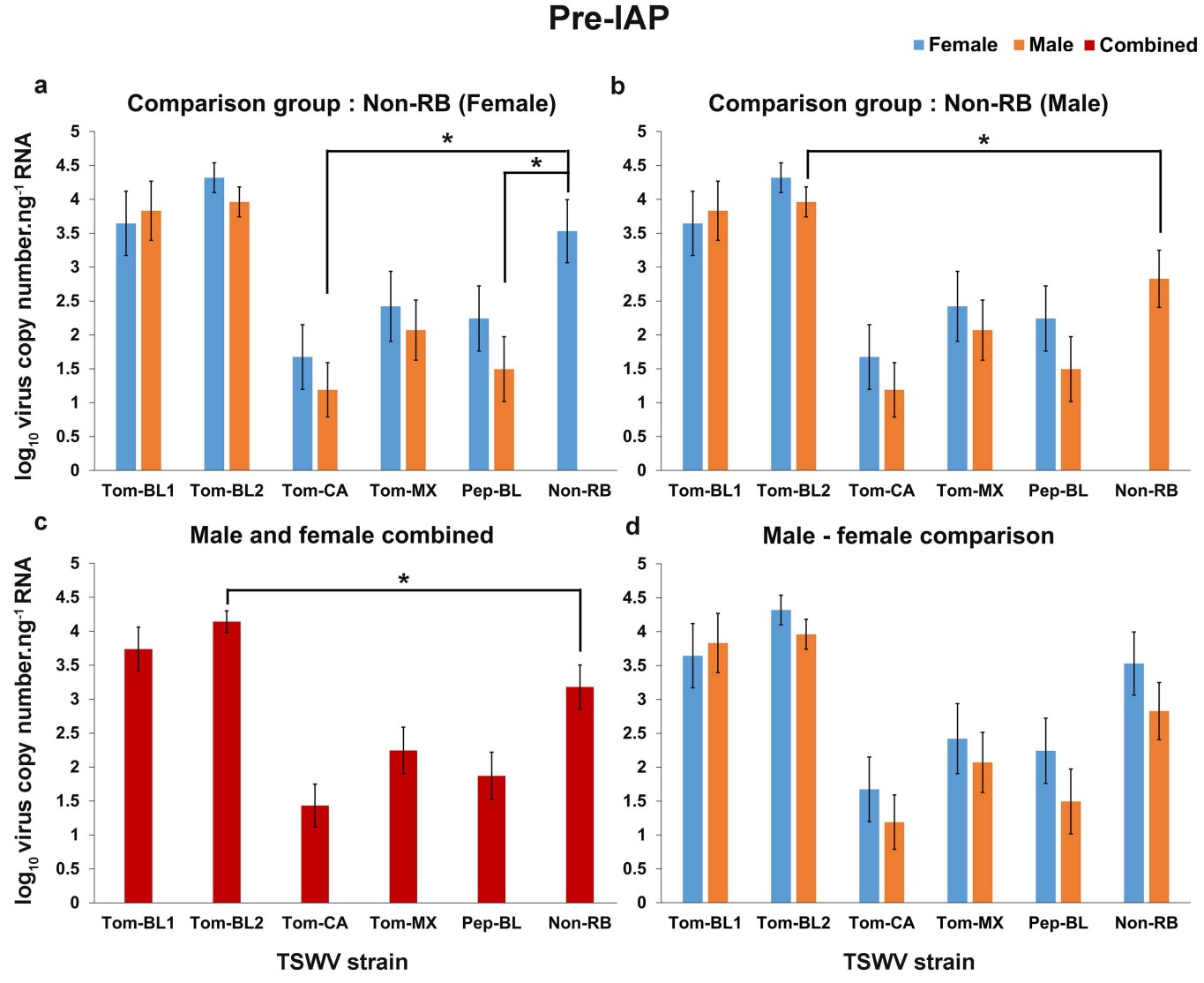

**Fig 3. TSWV copies.ng$^{-1}$ of RNA in adult *Frankliniella occidentalis* prior to inoculation access period.** Prior to IAP average copy number of different RB-TSWV strains carried by male and female *F. occidentalis* compared to Non-RB strain carried by either **(a)** female; or **(b)** male; (c) Prior to IAP average copy number of different strains harbored by *F. occidentalis* (male and female combined) compared to Non-RB strain carried by *F. occidentalis* (male and female combined); (d) Prior to IAP average copy number of different TSWV strains carried by female and male compared within the strains. Asterisks indicate significant differences at α = 0.05 (*$P < 0.05$, ** $P < 0.01$, *** $P < 0.001$).

**Post-IAP.** The highest and lowest virus copies were found in females of Tom-BL1 and Pep-BL strain, respectively and copy numbers in both treatments were significantly different from Non-RB strain copy numbers in females ($P < 0.01$, $P < 0.05$, Fig 4a) and males ($P < 0.05$, Fig 4b). When data were combined for males and females, the RB strain Tom-BL1 found to have higher virus copies; while Pep-BL had the lowest virus copies in thrips and were statistically different when compared to Non-RB strain ($P < 0.05$, Fig 4c). None of the other strains significantly differed from Non-RB strain (Fig 4c). No statistically significant difference was found between males and females carrying different TSWV strains (Fig 4d).

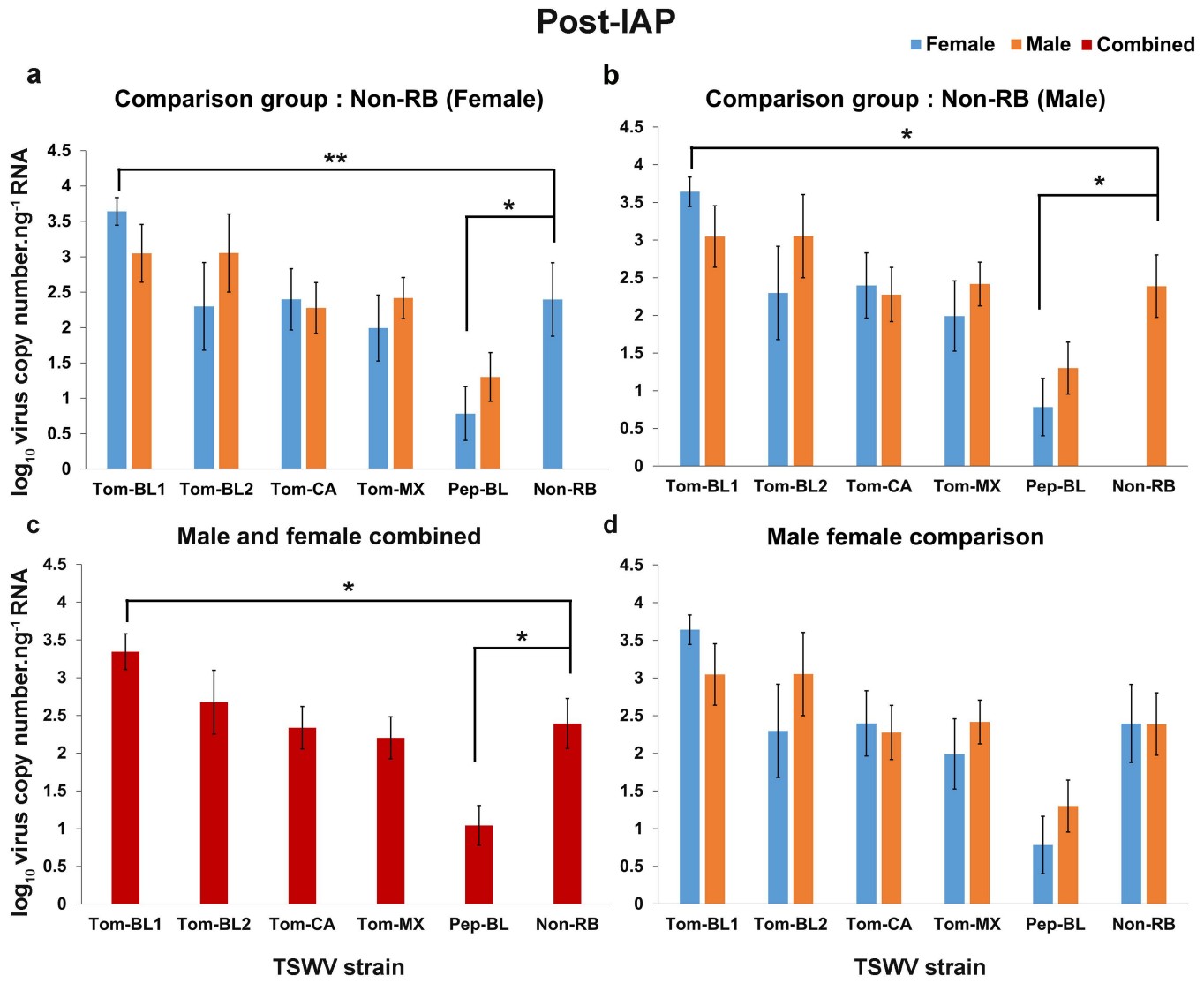

**Fig 4. TSWV copies.ng$^{-1}$ of RNA in adult *Frankliniella occidentalis* post inoculation access period.** Post-IAP average copy number of different RB-TSWV strains carried by male and female *F. occidentalis* compared to Non-RB strain carried by either **(a)** female; or **(b)** male; (c) Post-IAP average copy number of different strains harbored by *F. occidentalis* (male and female combined) compared to Non-RB strain carried by *F. occidentalis* (male and female combined); (d) Post-IAP average copy number of different TSWV strains carried by female and male compared within the strains. Asterisks indicate significant differences at α = 0.05 (*$P$ < 0.05, **$P$ < 0.01, ***$P$ < 0.001).

## Relationship of TSWV copies between source plant, insects, and inoculated leaf discs

Regression analysis revealed no correlation between virus copy numbers observed in female (F = 3.47; $P$ = 0.135; $R^2$ = 0.46) (Fig 5a) or male (F = 3.61; $P$ = 0.130; $R^2$ = 0.47) (Fig 5b) thrips and virus copies present in plant tissues that were used for virus acquisition. Similarly, there was no correlation between virus titers present in female (F = 1.27; $P$ = 0.321; $R^2$ = 0.24) (Fig 5c) or male (F = 2.50; $P$ = 0.188; $R^2$ = 0.38) (Fig 5d) thrips and virus copy numbers from thrips-inoculated leaf discs.

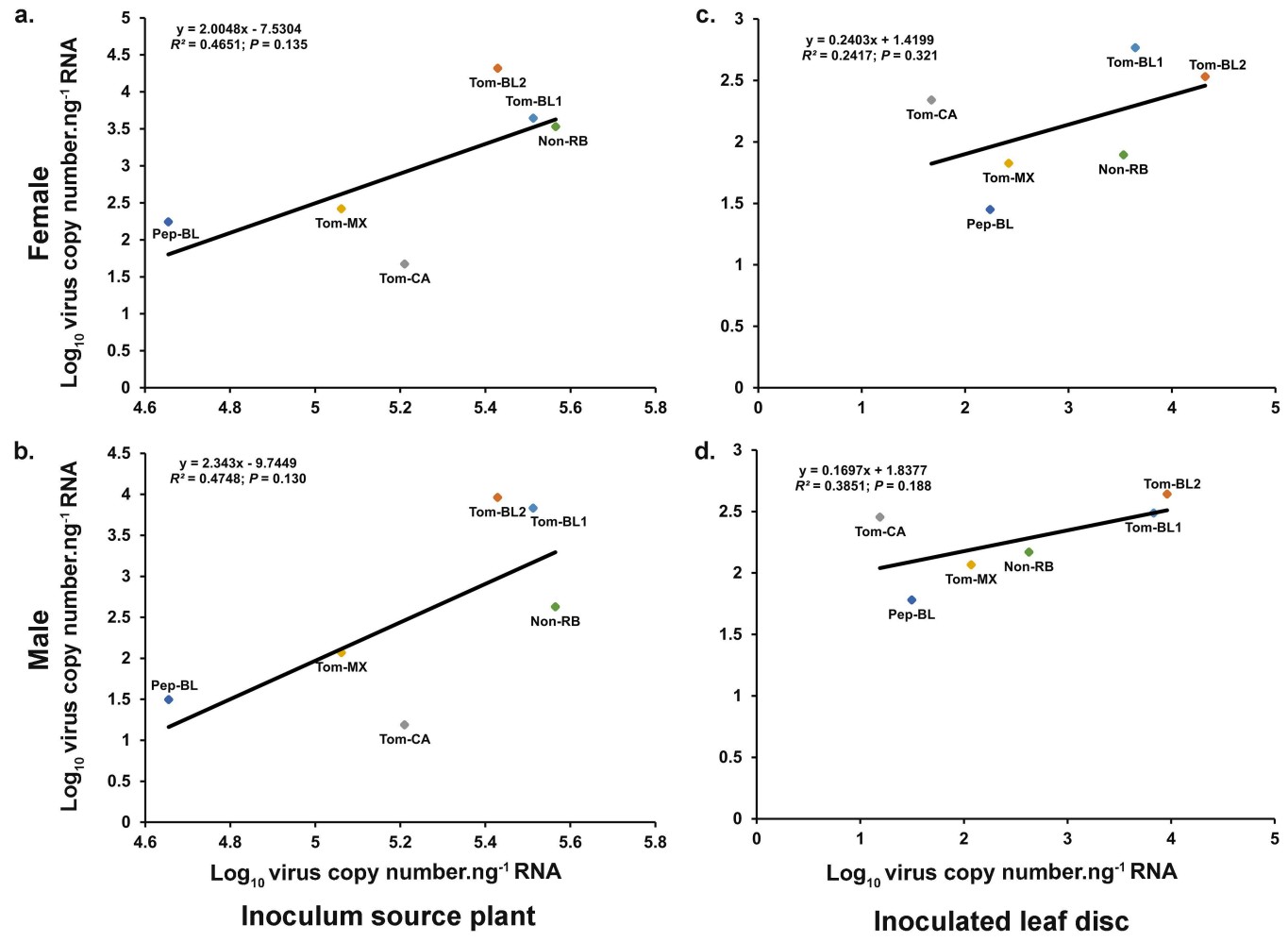

**Fig 5. Relationship between TSWV copies present in source plants, in viruliferous *Frankliniella occidentalis*, and inoculated leaf discs.** Regression line depicting a relationship between TSWV copies present in (a) source tissue and females and (b) source tissue and males. (c) inoculated leaf disc and female; (d) inoculated leaf disc and male thrips.

## TSWV inoculation efficiency of *F. occidentalis*

Overall, *F. occidentalis* inoculated TSWV at a higher efficiency at the first IAP (i.e., IAP-I) regardless of sex or strain. Percent inoculation decreased at the second IAP and the reduction was relatively higher for females compared to males, although no statistical difference was found. The inoculation efficiency remained low for III-IAP for most strains before it increased again for IAP-IV (Fig 6).

 *F. occidentalis* males inoculated Tom-BL1 (80% male vs 40% female) and Tom-CA (90% male vs 60% female) with higher efficiency than female at IAP-II (Fig 6a, 6c). However, at IAP-III, a steep increase in female efficiency over male was observed, which then eventually went up to 90–100% for both strains at IAP-IV. Tom-BL2 was inoculated with 100% efficiency by males during IAP-I and II. While the inoculation efficiency of females was substantially lower at IAP-II (50%) (Fig 6b), it went up to 90% at IAP-IV after reaching the same levels as males (30%) at IAP-III. Males inoculated Tom-MX at consistently higher rates across all four IAPs (90–100%), whereas females were relatively less efficient, particularly at

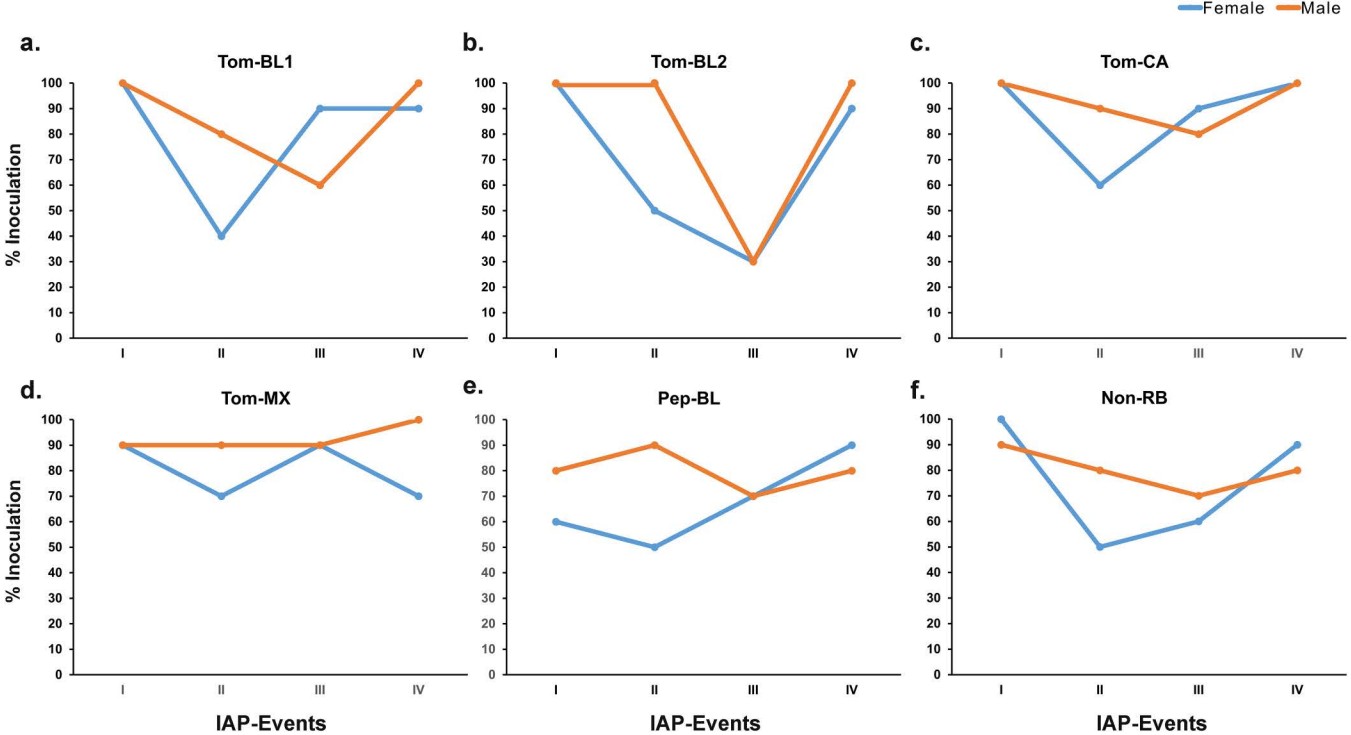

**Fig 6. Percent inoculation of different strains of TSWV by *Frankliniella occidentalis* over four consecutive IAPs.** Inoculation efficiency of female and male thrips to transmit (a) Tom-BL1; (b) Tom-BL2; (c) Tom-CA; (d) Tom-MX; (e) Pep-BL, and (f) Non-RB strains.

second and forth IAPs (70% each) ([Fig 6d]). Unlike other strains, Pep-BL inoculation at the first IAP was lower than 90% (80% male vs 60% female), which then increased by 10% in males and decreased by 10% in females at IAP-II before reaching 70% for both sexes. Despite the lower initial inoculation rate at the first IAP, females turned out to be more efficient than males at the fourth IAP (90% females vs 80% males) ([Fig 6e]). Both sexes inoculated Non-RB strain at 90–100% efficiency at IAP-I, after which efficiency dropped steeply in females to 50% vs 80% in males at IAP-II, followed by the recovery to 60% at IAP-III and 90% at IAP-III by females ([Fig 6f]).

Though there were differences in the inoculation of different strains across IAPs by males and females, no statistical differences were found. In most IAPs, male thrips inoculated virus either at a higher or similar rates to female thrips in all the RB strains, except for IAP-III or IV of Tom-CA, Pep-BL or Non-RB strains ([Fig 6]). Overall, the average % TSWV inoculation efficiency of adults at four IAPs infected with RB strains was 92, 72, 70, and 92%, while for Non-RB strain it was 95, 65, 65, and 85%, regardless of sex ([S5 Fig]). Furthermore, inoculation efficiency of females and males for RB strains was 77 and 87%, respectively, which was higher than that of females and males infected with Non-RB strain, 75 and 80%, respectively ([S6 Fig]). The inoculation efficiency in males surpassed that in females across the strains when all IAPs were combined. Furthermore, Tom-CA was inoculated with an efficiency of 92.5% by male thrips, while PepBL/Tom-BL2 had an inoculation efficiency of 67.5% by females, marking the highest and lowest inoculation efficiencies, respectively. ([S7 Fig]).

## Discussion

Over the years, *Sw-5b* and *Tsw* resistant tomato and pepper cultivars have provided resistance and served as the first line defense against TSWV [3,4]. However, the worldwide emergence of novel resistance breaking strains is rendering this single gene resistance only partly effective and, in some instances, non-effective [39]. Despite this, there are virtually

no studies investigating transmission biology of multiple strains in a greater detail. Building further on our prior work [31], we comprehensively assessed various aspects of transmission, namely, inoculation efficiency between sexes and virus accumulation of six novel strains of TSWV by western flower thrips, a predominant vector of TSWV. We found that TSWV is inoculated by WFT in all four IAPs. However, the efficiency of inoculation varied at each IAP and between sexes in a context-specific manner.

While thrips were able to consistently inoculate TSWV at each IAP, inoculation efficiency at each IAP was not consistent. A typical observed trend across most if not all strains was high inoculation at the first IAP, followed by a decline in the second, and finally the recovery in the third and fourth IAP. This is possibly due to a limited number of TSWV virions being available in the salivary vesicles [40] as a result of a lag to replenish new virus after the first IAP. The first IAP occurred immediately after adult eclosion which may be why the first IAP had higher inoculation efficiency. A window of 24-48h may be needed for TSWV to propagate in thrips, which plausibly led to increased efficiency at the third and fourth IAPs. This is consistent with a prior study by Van de wetering et al. [26] which reported that TSWV multiplies in adult thrips after their emergence and reached maximum titers in 4-day old thrips. The dynamics of virus titers before and after the inoculation access period (IAP) varied in a context-specific manner for Tom-BL1, Tom-BL2, and Pep-BL strains, but remained stable for the Tom-MX strain when compared to the Non-RB strain. Overall, the largely unchanged post-IAP virus titers across all strains indicate that virus levels do not significantly fluctuate after four successive inoculation events. This suggests that thrips, irrespective of sex, remain viruliferous throughout their lifespan—supporting earlier observations by Ullman et al. [41] and Nagata et al. [42].

Across all strains, male thrips were able to transmit TSWV higher than females, which is consistent with prior reports in this pathosystem [25,26]. It has been speculated that male feeding behavior, and higher mobility than females facilitate higher transmission rate. Females, on the contrary, due to their sedentary nature of feeding produce scars at the site of feeding i.e., necrotic spots, which prevent virus replication and impedes inoculation or transmission [43]. Interestingly, the TSWV silencing suppressor protein, NSs, which suppresses plant RNAi silencing machinery to facilitate TSWV infection, has been reported only in saliva of female thrips during feeding [44]. This was expected to facilitate TSWV inoculation by females better than males, but we didn't observe this in our study. This is possibly because mobility and feeding behavior were likely to be key contributors, more than NSs, in determining inoculation efficiency. Furthermore, mounting of plant defenses via RNAi is likely to have been compromised in leaf discs as opposed to entire plants.

The thrips were able to acquire all strains of TSWV, however, the numbers of virus particles accumulated in the thrips varied (Fig 5 and 6). Glycoproteins, Gn and Gc on the TSWV M segment are known to be involved in virus binding to thrip gut, before virions enter the cells probably via receptor mediated endocytosis [45,46]. The differences in interaction of virus strains, particularly their glycoproteins with TSWV-interacting proteins (TIPs) of the vector likely to have impacted TSWV propagation [47]. A recent study reported that infection of TSWV in thrips resulted in increased expression of a cytochrome P450 monooxygenases gene -CYP24, which suppressed the insect immune system and helped with the virus propagation. RNAi of NSs in thrips depleted the CYP24 transcripts while RNAi of N and NSm transcripts failed to reduce CYP24 expression, highlighting a potential role of NSs in suppressing thrip immune system [48]. Pep-BL, an originally pepper infecting strain, has several unique point mutations in the NSs compared to tomato infecting RB strains and Non-RB strains (unpublished data) which likely to impact Pep-BL accumulation in *F. occidentalis* since its titer was lower than other strains (Fig 3 and 4). These differences in virus titer in the adult thrips could be partly attributed to difference in the NSs protein.

Overall, when data from both female and male thrips were combined, the RB strains showed significantly higher inoculation efficiency than the Non-RB strain. While these differences were not statistically different, their biologically implications are not clear. Based on prior studies on source-sink relationship, virus titer in plants and in the insects were expected to impact virus acquisition and inoculation, respectively. However, we did not observe any correlation between virus titer in plant tissue used for acquisition (source), thrips, and thrips-inoculated tissues (sink). This suggests that regardless of the number of viral copies acquired by thrips, the virus propagation in the vector influences titer available for inoculation. However, a threshold for successful inoculation (minimum number of copies needed to be acquired for successful inoculation

and subsequent transmission) remain unknown. This could be further demonstrated by Pep-BL and Tom-MX strains. Although the number of virus particles inoculated by thrips were low, both strains were consistently inoculated at higher frequency (% inoculation) by thrips. This warrants a follow up study in which whether inoculation of leaf discs would translate into higher TSWV transmission in plants needs to be studied. Our prior study showed that RB strains increase the fitness of thrips compared to Non-RB strain. This follow up study suggests that this is likely due to the marginal increase in inoculation efficiency we observed in the present study. Additional factors include enhanced nutrient profiles and modified primary metabolism and defense response upon TSWV infection increasing vector fitness in terms of increased vector colonization and offspring [30].

For our experiments we used different plant cultivars as strains were maintained based on their resistance breaking ability in our laboratory. The virus inoculation experiment in the leaf disc lasted for only 24 hours as our focus was to determine inoculation efficiency of different strains by WFT. We believe that due to a short duration of experiment our results were not affected by genetic makeup of varieties. However, in future detailed studies can be conducted to compare acquisition and inoculation efficiency of WFT between different cultivars.

Transmission biology of TSWV RB strains, despite their worldwide emergence over the past decade, remains unexplored area of research. This work offers novel insights into thrips sex and inoculation parameters as potential determinants of TSWV RB transmission. However, more comprehensive investigations at transcriptional, protein and metabolic levels are needed to deepen our understanding of the intricate TSWV-thrips interactions and to devise tailored strategies for their management.

## Supporting information

**S1 Fig. TSWV copies.ng$^{-1}$ of RNA inoculated by *Frankliniella occidentalis* in IAP-I.** Average copy number of different TSWV strains transmitted by female (n = 10) and male (n = 10) thrips, and compared to Non-RB strain transmitted by either (a) female; or (b) male (c) average copy number of different strains inoculated by *F. occidentalis* (male and female combined) compared to Non-RB strain inoculated by *F. occidentalis* (male and female combined) (d) comparison of average copy number of different TSWV strains inoculated by female vs male within the strains. Asterisks indicate significant differences at $\alpha = 0.05$ (*$P < 0.05$, **$P < 0.01$, ***$P < 0.001$).
(TIF)

**S2 Fig. TSWV copies.ng$^{-1}$ of RNA inoculated by *Frankliniella occidentalis* in IAP-II.** Average copy number of different TSWV strains transmitted by female (n = 10) and male (n = 10) thrips, and compared to Non-RB strain transmitted by either (a) female; or (b) male (c) average copy number of different strains inoculated by *F. occidentalis* (male and female combined) compared to Non-RB strain inoculated by *F. occidentalis* (male and female combined) (d) comparison of average copy number of different TSWV strains inoculated by female vs male within the strains. Asterisks indicate significant differences at $\alpha = 0.05$ (* $P < 0.05$, ** $P < 0.01$, *** $P < 0.001$).
(TIF)

**S3 Fig. TSWV copies.ng$^{-1}$ of RNA inoculated by *Frankliniella occidentalis* in IAP-III.** Average copy number of different TSWV strains transmitted by female (n = 10) and male (n = 10) thrips, and compared to Non-RB strain transmitted by either (a) female; or (b) male (c) average copy number of different strains inoculated by *F. occidentalis* (male and female combined) compared to Non-RB strain inoculated by *F. occidentalis* (male and female combined) (d) comparison of average copy number of different TSWV strains inoculated by female vs male within the strains. Asterisks indicate significant differences at $\alpha = 0.05$ (*$P < 0.05$, **$P < 0.01$, ***$P < 0.001$).
(TIF)

**S4 Fig. TSWV copies.ng$^{-1}$ of RNA inoculated by *Frankliniella occidentalis* in IAP-IV.** Average copy number of different TSWV strains transmitted by female (n = 10) and male (n = 10) thrips, and compared to Non-RB strain transmitted by

either (a) female; or (b) male (c) average copy number of different strains inoculated by *F. occidentalis* (male and female combined) compared to Non-RB strain inoculated by *F. occidentalis* (male and female combined) (d) comparison of average copy number of different TSWV strains inoculated by female vs male within the strains. Asterisks indicate significant differences at α = 0.05 (*$P < 0.05$, **$P < 0.01$, ***$P < 0.001$).
(TIF)

**S5 Fig. Inoculation efficiency of RB strains in four IAPs by *Frankliniella occidentalis* (TomBL1, Tom-BL2, Tom-CA, Tom-MX, and Pep-BL2 combined) compared with Non-.RB.**
(TIF)

**S6 Fig. Overall percent inoculation efficiency of RB strains (TomBL1, Tom-BL2, Tom-CA, Tom-MX, and Pep-BL2 combined) compared to Non-RB strains by female and male *Frankliniella occidentalis*.**
(TIF)

**S7 Fig. Percent inoculation efficiency of different strains by male and females of *Frankliniella occidentalis* after combining all the events.**
(TIF)

## Acknowledgments

We thank undergraduate researchers Justice Crowder, Cayla Moore, and Jewels Hernandez for their help with thrips colony maintenance.

## Author contributions

Conceptualization: Kiran R. Gadhave.

Data curation: Senthilraja Chinnaiah, Arinder K. Arora.

Formal analysis: Senthilraja Chinnaiah, Arinder K. Arora.

Funding acquisition: Kiran R. Gadhave.

Investigation: Senthilraja Chinnaiah.

Methodology: Senthilraja Chinnaiah.

Project administration: Kiran R. Gadhave.

Resources: Kiran R. Gadhave.

Software: Arinder K. Arora.

Supervision: Kiran R. Gadhave.

Validation: Arinder K. Arora.

Visualization: Senthilraja Chinnaiah, Arinder K. Arora.

Writing – original draft: Senthilraja Chinnaiah.

Writing – review & editing: Senthilraja Chinnaiah, Arinder K. Arora, Kiran R. Gadhave.

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
