## [Decision Letter · Decision Letter 0]

Dear Dr. Gadhave,

We look forward to receiving your revised manuscript.

Kind regards,

Sumit Jangra, Ph.D.

Academic Editor

PLOS ONE

“Texas A&M AgriLife Research Insect Vectored Diseases Grant”

Reviewers' comments:

Reviewer's Responses to Questions

**Comments to the Author**

1. Is the manuscript technically sound, and do the data support the conclusions?

Reviewer #1: No

Reviewer #2: Yes

Reviewer #3: Yes

2. Has the statistical analysis been performed appropriately and rigorously?

Reviewer #1: No

Reviewer #2: Yes

Reviewer #3: Yes

3. Have the authors made all data underlying the findings in their manuscript fully available?

Reviewer #1: Yes

Reviewer #2: Yes

Reviewer #3: Yes

4. Is the manuscript presented in an intelligible fashion and written in standard English?

Reviewer #1: No

Reviewer #2: Yes

Reviewer #3: Yes

Reviewer #1: The manuscript “Novel strains of tomato spotted wilt orthotospovirus (TSWV) are transmitted by western flower thrips in a context-specific manner” reports on the differences of thrips (Frankliniella occidentalis - WFT) transmission efficiencies among several resistance-breaking (RB) TSWV strains. This study follows a previous investigation reporting positive effects of the same RB strains on the WFT fitness. This new study provides evidence that WFT transmits some RB strains at a higher rate than non-RB strains, and WFT males showed to be more efficient vectors than females.

The study deals with the understudied role of thrips vectors in spreading TSWV RB strains, but the obtained results are not so outstanding to be considered for publication on PLOS ONE. Also, the paper needs an extensive revision of both English language and data organisation. I would suggest the authors to re-write the paper and to consider its publication on a journal which is limited to the field of plant viruses.

INTRODUCTION:

Line 41: replace “produce” with “production”.

Lines 45-48: the sentences are redundant, re-write a unique sentence.

Line 50: “TSWV is transmitted in a persistent…”

Line 52: “…is considered the most effective vector due to its high reproductive rate as well as concealed and polyphagous behaviour.”.

Line 55: replace “infected” with “viruliferous”.

Lines 57-58: rephrase the sentence, the English is poor. Replace “transmitters” with “vectors”.

Line 58: “was attributed to their mobility, feeding behaviours, and less leaf scar productions”: explain how these parameters can influence the transmission of TSWV.

Lines 62-63: explain how TSWV influences the insect behaviour. The same for the aminoacid content: how the virus can affect the aa content? Also, some of the references 27-31 do not concern WFT: please check.

Lines 65-68: please, write a unique sentence and improve the English language. What does “adult period” mean? How did you measure it?

Line 69: “RB-viruliferous thrips transmitted TSWV more efficiently than the Non-RB strains”.

Lines 70-72: this sentence should be moved earlier in the ms: it is not the first time you mention the RB strains.

Lines 75-83: both English language and content should be improved, it is very hard to understand the meaning of the paragraph. For example, the sentence: “differences in inoculation rates of males and females to transmit different strains” does not make any sense. Also: which is the difference between point (i) and (iii)?

MATERIALS AND METHODS

Line 98: Replace “Of” with “Among the”.

Line 99: typical and characteristic: redundant.

Line 103-4: The sentence is not correct, rephrase.

Line 111: remove “in previous study”.

Line 135: inoculate the virus

Line 141-2: it is not clear what did you stored: the leaf discs, the water???? Rephrase

Lines 150-5: what about the volumes/concentrations of reagents? What about the PCR conditions?

RESULTS

Line 177: rephrase the title: you cannot inoculate a virus titre and insects do not inoculate viruses but they transmit viruses.

The Results section and the figures are very confusing. It is difficult to read and follow the description of the experimental results and to catch the final aim of the study (which should be the assessment of differences between the transmission features of RB and not-RB TSWV strains, if I have understood well….).

I strongly suggest to re-organise the section and to perform the statistical analyses differently. For example, I would suggest to report the main results in the following order:

1. Acquisition efficiency = virus titre in WFT before IAP. First, assess if there are any differences between male and female within each strain. Second, cumulate data of female and males for each RB strain, and for each RB strain assess if the virus titre is significantly lower/higher than non-RB strain.

2. Transmission efficiency = virus titre in the inoculated plants at 4 IAPs. First, assess if there are any differences between male and females within each strain and then cumulate the data when you can. Secondly, compare the virus titre of each RB strain with that of non-RB strain. Third, highlight at which IAP you measured the highest titres and the biggest differences.

3. Virus titre in WFT after IAP: why did you perform this analysis? Which is the aim? Did you want to measure the replication rates of the different virus strains within the vector? I do not think is related to the rest of the study. As well, I cannot see any comment on this in the Discussion section.

Figures: when you are comparing several items in the same graph, use letters (a,b,c etc…) instead of asterisks to indicate significant differences.

Figure 2: the figure lacks the most important result of this study: the significant differences in transmission rates between each RB strain and the non-RB strain.

DISCUSSION:

The Discussion needs to be deeply revised considering the changes I suggested in the Results.

I just highlight two main things that appeared wrong to me:

Lines 373-4: the sentence is not true for all the RB strains.

Lines 383-4: the sentence is contradictory.

Reviewer #2: All the experiments have been conducted in a meticulous manner. The manuscript has been well written. There are very few minor corrections. It may be amended for further improvement.

Line 19-23: Kindly break the sentence and rewrite for better understanding.

Line 24: Inoculated ? or transmitted? Kindly rewrite the sentence.

Line 25: Inoculation efficiency ? or transmission efficiency? Kindly revise through out the text.

Line 39-41: Kindly break the sentence and rewrite for better understanding.

Line 48: United states

Line 50: Modify “TSWV is transmitted by plant-feeding thrips in a persistent and propagative manner”

Line 70: The authors have mentioned the use of isolate instead of strain. However, there are some side heads with the word “strain” Kindly clarify. Modify if necessary.

Line 338: Correct “possibly”

Line 353: “…saliva of the female thrips”

Line 361-363: Rewrite the sentence for better understanding.

The discussion part may be strengthened more.

Reviewer #3: This manuscript addresses an important and timely topic—understanding the transmission biology of resistance-breaking (RB) strains of Tomato spotted wilt virus (TSWV) by Frankliniella occidentalis (Western Flower Thrips). The current work provides insights into inoculation efficiency, virus titers, and sex-specific transmission differences between RB and non-RB strains. The study is well-structured, and the results and objectives are clearly presented. It is mostly in good shape for publication and no further experiments are needed; however, a minor revision is required to improve the quality of the current version.

Minor revision:

98 Provide reference for the TSWV strains used in this study

107 What is the significance of using consecutive IAPs of 24h? How would the outcome vary if the IAPs were 24h, 48h, 72h, and 96h post-eclosion? How would the outcome vary between these scenarios? In the field conditions, transmission occurs in both the scenarios. Add a note in the discussion section.

121 What leaf disks were used? Are they different for tomato and pepper strains? Were they pre-tested for TSWV?

132 Provide reference for insect morphometric characters.

195 Please correct this statement. Virus copy number of Tom-BL2 inoculated by females and males was significantly higher in IAP-I than other IAP-I & II (P<0.001; Fig. 2b) and IAP-III (P<0.05; Fig.2b).

363. The word- plausible is used frequently. Consider using alternative terms to improve readability.

375 The regression analysis reveals no correlation between virus titer in plant tissue used for acquisition (source), thrips, and thrips-inoculated tissues (sink). Have you considered other factors, such as the spatial distribution of the virus, tissue heterogeneity, and sensitivity of virus quantification? Please explain this in detail.

The URLs for some references are leading to a different source. For example: reference 31 and 32. Please verify and correct these, check the remaining references for accuracy.

**Do you want your identity to be public for this peer review?** For information about this choice, including consent withdrawal, please see our Privacy Policy

Reviewer #1: No

Reviewer #2: No

Reviewer #3: **Yes: ** Kishorekumar Reddy

---

## [Author Response · Author response to Decision Letter 1]

16 May 2025

Dr Sumit Jangra, Ph.D.

Academic Editor

PLOS One

May 15, 2025

Dear Dr Jangra,

We sincerely thank you and reviewers for their thoughtful, constructive, and insightful feedback, which has substantially improved the quality and clarity of our manuscript: Novel strains of tomato spotted wilt orthotospovirus (TSWV) are transmitted by western flower thrips in a context-specific manner. We have carefully addressed each of the concerns raised, including clarifications of the text, detailed explanations of the experimental design, and justifications for our analytical approach. Our point-by-point responses are provided below, with revisions indicated in blue. We greatly appreciate your time, effort, and valuable input throughout this review process.

In this study, we conducted an in-depth investigation into the transmission biology of tomato spotted wilt virus (TSWV), a globally significant agricultural pathogen, and its primary vector, the Western flower thrips, a widespread and damaging pest. The novelty of our work lies in two key aspects. First, this is only the second study—building upon our previous research—to examine the transmission biology of novel resistance-breaking strains of TSWV, which have emerged globally in recent years. Second, we delve into how these novel strains are transmitted by thrips in a context-specific manner, focusing on three critical determinants of transmission: TSWV inoculation efficiency, the sex of the thrips and source-sink relationships between virus titers.

We believe this manuscript aligns well with the journal's interdisciplinary scope, as it examines the intricate vector-virus interactions within a globally significant pathosystem. Given that TSWV is among the top ten most economically significant plant viruses worldwide, our findings are likely to attract the interest of a broad, interdisciplinary audience and contribute to advancing research in this field. The paper is not currently being considered for publication elsewhere. All authors have contributed to its production, and all have agreed on the final version. We hereby declare that there is no conflict of interest for any of the authors.

This research was funded by Texas A&M AgriLife Research Insect Vectored Diseases Grant. The funder had no role in study design, data collection and analysis, decision to publish, or preparation of the manuscript.

Response to reviewer 1:

Reviewer #1: The manuscript “Novel strains of tomato spotted wilt orthotospovirus (TSWV) are transmitted by western flower thrips in a context-specific manner” reports on the differences of thrips (Frankliniella occidentalis - WFT) transmission efficiencies among several resistance-breaking (RB) TSWV strains. This study follows a previous investigation reporting positive effects of the same RB strains on the WFT fitness. This new study provides evidence that WFT transmits some RB strains at a higher rate than non-RB strains, and WFT males showed to be more efficient vectors than females.

The study deals with the understudied role of thrips vectors in spreading TSWV RB strains, but the obtained results are not so outstanding to be considered for publication on PLOS ONE. Also, the paper needs an extensive revision of both English language and data organisation. I would suggest the authors to re-write the paper and to consider its publication on a journal which is limited to the field of plant viruses.

We thank Reviewer #1 for their thoughtful comments and constructive feedback. While we fully agree that the role of thrips in disseminating RB strains remains largely unexplored, we respectfully differ with their opinion regarding the manuscript’s suitability for publication in PLOS ONE. Consistent with the perspectives of Reviewers #2 and #3, we believe that the novelty of our findings, the interdisciplinary nature of the work spanning entomology, virology, and molecular biology, and the broad scientific scope of PLOS ONE make this manuscript well-suited for the journal. We have addressed all of Reviewer #1’s comments in a detailed, point-by-point manner. Where our interpretation differs, we have provided a clear and well-supported rationale, aligning with the constructive perspectives shared by Reviewers #2 and #3. We have taken great care in preparing this manuscript, and we appreciate the recognition of its clarity and rigor by the other reviewers. That said, all reviewer comments have been carefully considered, and we have made additional revisions to further enhance the manuscript’s clarity. A detailed explanation of our data organization strategy follows below.

INTRODUCTION:

Line 41: replace “produce” with “production”.

L41 The “produce” in this context is referred to agricultural produce such as fruits, vegetables etc. grown and sold for human consumption. “Yield and quality of production” would change the meaning as production is referred to the process of production rather than the end produce. To bring the clarity, the word “agricultural” is added.

Lines 45-48: the sentences are redundant, re-write a unique sentence.

L45-48 The sentence is rewritten.

Line 50: “TSWV is transmitted in a persistent…”

L49: Text is rephrased

Line 52: “…is considered the most effective vector due to its high reproductive rate as well as concealed and polyphagous behaviour.”

L51-52. Text is rephrased as suggested.

Line 55: replace “infected” with “viruliferous”.

L55: Text is rephrased

Lines 57-58: rephrase the sentence, the English is poor. Replace “transmitters” with “vectors”.

L56-58: Text is rephrased to improve clarity.

Line 58: “was attributed to their mobility, feeding behaviours, and less leaf scar productions”: explain how these parameters can influence the transmission of TSWV.

L56-59. An explanation of this is now provided in the revised text.

Lines 62-63: explain how TSWV influences the insect behaviour. The same for the amino acid content: how the virus can affect the aa content? Also, some of the references 27-31 do not concern WFT: please check.

Previous studies have shown that TSWV infection, both directly in the vector and indirectly through the plant host, enhances vector fitness by increasing behavior activity, fecundity, and longevity. For instance, Maris et al. (2004) reported that TSWV-infected plants were more attractive to western flower thrips (WFT) and led to higher offspring numbers. In terms of indirect effects, TSWV-infected plants were found to contain up to 15 times more free amino acids compared to healthy plants, potentially boosting egg production (Shrestha et al., 2012). Additionally, TSWV-infected WFT exhibited increased longevity and survival, and infected leaf discs were more attractive than healthy ones (Ogada et al., 2012). Furthermore, Nachappa et al. (2020) reported that TSWV infection enhances vector fitness by modulating host plant metabolic and defence pathways. All of these references are specific to thrips/TSWV pathosystem and have therefore been cited in the text (27-31). A description is slightly revised to improve clarity (L64).

Lines 65-68: please, write a unique sentence and improve the English language. What does “adult period” mean? How did you measure it?

“Adult period” is a commonly used entomological term refers to a period in days from the first day of adulthood through the death of adult insect. We reared adult thrips on clean bean plants from the first day of adulthood until their death, recording the number of days each adult lived (n = 10). This term is clearly defined in the text (L68-69).

Line 69: “RB-viruliferous thrips transmitted TSWV more efficiently than the Non-RB strains”.

L70: The sentence is rephrased.

Lines 70-72: this sentence should be moved earlier in the ms: it is not the first time you mention the RB strains.

L71. This is the first mention of the term 'strain' in regard to our strains in the main body of the manuscript. So, we believe this sentence is appropriately placed.

Lines 75-83: both English language and content should be improved, it is very hard to understand the meaning of the paragraph. For example, the sentence: “differences in inoculation rates of males and females to transmit different strains” does not make any sense. Also: which is the difference between point (i) and (iii)?

L76-83. We’ve restructured this paragraph to bring more clarity.

MATERIALS AND METHODS

Line 98: Replace “Of” with “Among the”.

L98. Text is rephrased as suggested.

Line 99: typical and characteristic: redundant.

L99. Only ‘typical’ is retained as suggested.

Line 103-4: The sentence is not correct, rephrase.

L103-105. Text is rephrased as suggested.

Line 111: remove “in previous study”.

L111. Text is removed as suggested.

Line 135: inoculate the virus

L134. Text is rephrased as suggested.

Line 141-2: it is not clear what did you stored: the leaf discs, the water???? Rephrase

L141. Text is rephrased as suggested.

Lines 150-5: what about the volumes/concentrations of reagents? What about the PCR conditions?

L150-157: These details had been provided in our prior publication Chinnaiah et al. (2023). However, they have been included in the text.

RESULTS

Line 177: rephrase the title: you cannot inoculate a virus titre and insects do not inoculate viruses but they transmit viruses.

Line 178: The text has been rephrased in alignment with Reviewer #2’s comments. We have been deliberate in our use of terminology to ensure accurate representation of our findings and to minimize the potential for misinterpretation. Specifically, we use the term “inoculation efficiency” rather than “transmission efficiency” because the thrips inoculated virus into leaf discs, not whole plants—a distinction critical to our experimental design, which focused on quantifying virus titers. This should not be conflated with the broader term “transmission biology”, which we use to refer to the full range of parameters assessed in this study, including inoculation efficiency, sex-specific differences in inoculation, virus accumulation, and source-sink relationships. These terminological distinctions have been applied consistently throughout the revised manuscript.

The Results section and the figures are very confusing. It is difficult to read and follow the description of the experimental results and to catch the final aim of the study (which should be the assessment of differences between the transmission features of RB and not-RB TSWV strains, if I have understood well….).

I strongly suggest to re-organise the section and to perform the statistical analyses differently. For example, I would suggest to report the main results in the following order:

1. Acquisition efficiency = virus titre in WFT before IAP. First, assess if there are any differences between male and female within each strain. Second, cumulate data of female and males for each RB strain, and for each RB strain assess if the virus titre is significantly lower/higher than non-RB strain.

The objective of our study, as clearly outlined in the Abstract and Introduction, was to investigate multiple transmission parameters—namely, inoculation efficiency, putative sex-specific differences in transmission, virus accumulation, and source–sink relationships—to better understand the transmission biology of RB and Non-RB strains. With this framework, we conducted a comprehensive analysis and presented the complex dataset in a clear, stepwise fashion across multiple figures. For example, in Figure 3, we examined virus titers of different strains acquired by each sex and compared them to the Non-RB strain acquired by females (Fig. 3a) and males (Fig. 3b). We further compared cumulative virus titers between RB and Non-RB strains by pooling data from both sexes (Fig. 3c), and analyzed sex-specific differences within each strain (Fig. 3d). We believe this sequential and logical data presentation enhances clarity and accessibility for readers—a sentiment echoed by Reviewers #2 and #3. To further aid navigation, Figure 1 provides a methodological overview; Figure 2 shows virus titers in leaf discs across four inoculation events; Figures 3 and 4 present pre- and post-IAP virus titers in thrips; Figure 5 illustrates source–sink relationships; and Figure 6 details inoculation efficiency dynamics across events and between sexes.

2. Transmission efficiency = virus titre in the inoculated plants at 4 IAPs. First, assess if there are any differences between male and females within each strain and then cumulate the data when you can. Secondly, compare the virus titre of each RB strain with that of non-RB strain. Third, highlight at which IAP you measured the highest titres and the biggest differences.

We defined inoculation efficiency based on the percentage of leaf discs successfully inoculated, not on virus titers (see revised Introduction for clarification). This parameter is clearly presented as percent inoculation efficiency in Figure 6. To maintain readability and avoid overwhelming the main text with excessive data, we have provided extensive supporting information in the supplementary files for readers who wish to explore the dataset in greater depth. For example, Supplementary Figure 5 presents percent inoculation data pooled across all strains for males and females, compared across different IAPs. In Supplementary Figure 7, we combined all IAPs to compare percent inoculation efficiency between sexes across strains. Additionally, we analyzed virus titers transmitted during each IAP: Supplementary Figures 1–4 show differences in virus copy numbers inoculated by males versus females during each IAP, along with cumulative analyses by sex for each inoculation event.

3. Virus titre in WFT after IAP: why did you perform this analysis? Which is the aim? Did you want to measure the replication rates of the different virus strains within the vector? I do not think is related to the rest of the study. As well, I cannot see any comment on this in the Discussion section.

Viruliferous WFT transmit TSWV throughout their lifespan, as the virus replicates within the vector. In our study, we quantified virus titers post-IAP to assess whether WFT remain viruliferous after four consecutive transmission events, and whether there is a substantial reduction in virus titers over time. We thank Reviewer #1 for pointing out the absence of a brief discussion on this aspect. We have now addressed this in the revised manuscript (lines 345–351).

Figures: when you are comparing several items in the same graph, use letters (a,b,c etc…) instead of asterisks to indicate significant differences.

We intentionally used asterisks (*, **, ***) rather than letter annotations (a, b, c), as the asterisks allow us to indicate specific levels of statistical significance—p < 0.05, p < 0.01, and p < 0.001, respectively. This level of detail cannot be conveyed as precisely using letter-based groupings.

Figure 2: the figure lacks the most important result of this study: the significant differences in transmission rates between each RB strain and the non-RB strain.

This information is provided in Supplementary figures 1-4 due to the intricate nature of this data. The key findings from these figures have been presented in Results section.

DISCUSSION:

The Discussion needs to be deeply revised considering the changes I suggested in the Results.

The missing text on Post-IAP in discussion as suggested by Reviewer 1 is added.

I just highlight two main things that appeared wrong to me:

Lines 373-4: the sentence is not true for all the RB strains.

L379-380: The text is rephrased as suggested.

Lines 383-4: the sentence is contradictory.

L390-391: The text is modified for clarity.

Reviewer #2: All the experiments have been conducted in a meticulous manner. The manuscript has been well written. There are very few minor corrections. It may be amended for further improvement.

We sincerely thank Reviewer #2 for their compliments and constructive feedback. Their thoughtful assessment and supportive remarks are much appreciated. We have carefully addressed the minor corrections to further improve the manuscript.

Line 19-23: Kindly break the sentence and rewrite for better understanding.

L21: Sentence is split into two for better understandin

---

## [Decision Letter · Decision Letter 1]

Dear Dr. Gadhave,

Thank you for submitting your manuscript to PLOS ONE. After careful consideration, we feel that it has merit but does not fully meet PLOS ONE’s publication criteria as it currently stands. Therefore, we invite you to submit a revised version of the manuscript that addresses the points raised during the review process.

We look forward to receiving your revised manuscript.

Kind regards,

Sumit Jangra, Ph.D.

Academic Editor

PLOS ONE

Journal Requirements:

Reviewers' comments:

Reviewer's Responses to Questions

**Comments to the Author**

Reviewer #1: (No Response)

Reviewer #3: All comments have been addressed

2. Is the manuscript technically sound, and do the data support the conclusions?

Reviewer #1: Partly

Reviewer #3: Yes

3. Has the statistical analysis been performed appropriately and rigorously?

Reviewer #1: No

Reviewer #3: Yes

4. Have the authors made all data underlying the findings in their manuscript fully available?

Reviewer #1: Yes

Reviewer #3: Yes

5. Is the manuscript presented in an intelligible fashion and written in standard English?

Reviewer #1: Yes

Reviewer #3: Yes

Reviewer #1: The manuscript “Novel strains of tomato spotted wilt orthotospovirus (TSWV) are transmitted by western flower thrips in a context-specific manner” reports on the differences of thrips (Frankliniella occidentalis - WFT) transmission efficiencies among several resistance-breaking (RB) TSWV strains. This study follows a previous investigation reporting positive effects of the same RB strains on the WFT fitness. This new study provides evidence that WFT transmits some RB strains at a higher rate than non-RB strains, and WFT males showed to be more efficient vectors than females.

The authors addressed most of the issues raised in the first version of the manuscript. I keep my opinion that the inoculation efficiency analysed for different RB strains, male vs female thrips, and different IAPs should be firstly put in correlation with the acquisition efficiency: if the thrips differently acquired TSWV at the above-mentioned conditions this would influence the results of the inoculation efficiency experiments. I cannot see any consideration in Results, Discussion, and Figure 2 about the comparison of pre- and post-IAP results.

I still have some minor comments:

INTRODUCTION:

Line 41: to the best ok my knowledge, “produce” is a verb not a noun. The authors should replace it with a noun, like product or production.

References 27-31: not all the references refer to F. occidentalis whereas the sentence in the text refers to this species only. This should be solved.

Line 78: “where viruliferous WFT feed on a series of leaf discs”

Line 103: “The non-RB strain, which is unable to overcome resistance conferred by Sw-5b or Tsw,”

Figures: you may use the letter annotation and indicate the level of significance into the legend, this would simplify the readability of the figures.

Line 382: the sentence: “Overall, when the females and males combined, RB strains inoculated with higher efficiency than the Non-RB stain.” still lacks the meaning.

Reviewer #3: (No Response)

**Do you want your identity to be public for this peer review?** For information about this choice, including consent withdrawal, please see our Privacy Policy

Reviewer #1: No

Reviewer #3: **Yes: ** Kishorekumar Reddy

---

## [Author Response · Author response to Decision Letter 2]

18 Jun 2025

Response to reviewer 1:

The manuscript “Novel strains of tomato spotted wilt orthotospovirus (TSWV) are transmitted by western flower thrips in a context-specific manner” reports on the differences of thrips (Frankliniella occidentalis - WFT) transmission efficiencies among several resistance-breaking (RB) TSWV strains. This study follows a previous investigation reporting positive effects of the same RB strains on the WFT fitness. This new study provides evidence that WFT transmits some RB strains at a higher rate than non-RB strains, and WFT males showed to be more efficient vectors than females.

The authors addressed most of the issues raised in the first version of the manuscript. I keep my opinion that the inoculation efficiency analysed for different RB strains, male vs female thrips, and different IAPs should be firstly put in correlation with the acquisition efficiency: if the thrips differently acquired TSWV at the above-mentioned conditions this would influence the results of the inoculation efficiency experiments. I cannot see any consideration in Results, Discussion, and Figure 2 about the comparison of pre- and post-IAP results.

We thank Reviewer 1 for recognizing the novel aspects of our study and for acknowledging how prior comments were addressed. We agree that acquisition efficiency is a critical factor influencing inoculation outcomes. To minimize variability in virus acquisition, we used synchronized cohorts of thrips larvae (both male and female), all subjected to identical acquisition access periods (AAPs) on the same infected leaf tissue.

While acquisition efficiency was measured separately for each group and strain, our primary objective was to assess relative inoculation efficiency between sexes and IAP events within each strain under standardized acquisition conditions. Thus, any observed differences in inoculation efficiency across RB strains, sexes, or IAP events likely reflect post-acquisition dynamics rather than differences in acquisition itself.

For these reasons, we present the data from an “inoculation-centric” perspective, as it directly reflects the central step in transmission, while still accounting for acquisition in a controlled and robust manner.

INTRODUCTION:

Line 41: to the best ok my knowledge, “produce” is a verb not a noun. The authors should replace it with a noun, like product or production.

L41: The word “produce” is replaced with “product”

References 27-31: not all the references refer to F. occidentalis whereas the sentence in the text refers to this species only. This should be solved.

L64. Following this suggestion, the previous reference number 29, which was the only one referring to F. fusca, has been removed.

Line 78: “where viruliferous WFT feed on a series of leaf discs”

L78. The text is rephrased.

Line 103: “The non-RB strain, which is unable to overcome resistance conferred by Sw-5b or Tsw,”

L103-104. The text is rephrased as suggested.

Figures: you may use the letter annotation and indicate the level of significance into the legend, this would simplify the readability of the figures.

We sincerely appreciate the reviewer’s suggestion. However, we believe the current figures effectively convey the significant differences between treatments and their levels in a clear and accessible manner. Including both letter annotations within the figures and significance levels in the legend may compromise overall readability and visual clarity. We have aimed to strike a balance between detail and interpretability and feel the current format best serves that purpose.

Line 382: the sentence: “Overall, when the females and males combined, RB strains inoculated with higher efficiency than the Non-RB stain.” still lacks the meaning.

L382-383. The text has been revised for more clarity.

We sincerely thank Reviewer #3, Dr. Kishorekumar Reddy, for recommending our manuscript for publication.

---

## [Editor Report · Decision Letter 2]

Novel strains of tomato spotted wilt orthotospovirus (TSWV) are transmitted by western flower thrips in a context-specific manner

PONE-D-25-17574R2

Dear Dr. Gadhave,

We’re pleased to inform you that your manuscript has been judged scientifically suitable for publication and will be formally accepted for publication once it meets all outstanding technical requirements.

Kind regards,

Sumit Jangra, Ph.D.

Academic Editor

PLOS ONE
---

## [Editor Report · Acceptance letter]

PONE-D-25-17574R2

PLOS ONE

Dear Dr. Gadhave,

I'm pleased to inform you that your manuscript has been deemed suitable for publication in PLOS ONE. Congratulations! Your manuscript is now being handed over to our production team.

Kind regards,

on behalf of

Dr. Sumit Jangra

Academic Editor

PLOS ONE